



# 1 Measurement report: Per- and polyfluoroalkyl substances (PFAS) in
# 2 particulate matter (PM₁₀) from activated sludge aeration

Jishnu Pandamkulangara Kizhakkethil[1], Zongbo Shi[2], Anna Bogush[1], and Ivan Kourtchev[1]
[1]Centre for Agroecology Water and Resilience (CAWR), Coventry University, Wolston Lane, Ryton on Dunsmore CV8 3LG,
UK
[2]School of Geography, Earth and Environmental Sciences, University of Birmingham, Edgbaston, Birmingham, B15 2TT, UK
*Correspondence to*: Ivan Kourtchev (ivan.kourtchev@coventry.ac.uk)
**Abstract.** Environmental pollution with per- and polyfluoroalkyl substances (PFAS), commonly referred to as "forever
chemicals", received significant attention due to their environmental persistence and bioaccumulation tendencies. Effluents
from wastewater treatment plants (WWTPs) have been reported to contain significant levels of PFAS. Wastewater treatment
processes such as aeration have the potential to transfer PFAS into the atmosphere. However, understanding their fate during
sewage treatment remains challenging. This study aims to assess aerosolisation of PFAS during WWTP process. Special
emphasis is given to new generation and legacy PFAS (e.g., perfluorooctanesulfonic acid (PFOS) and perfluorooctanoic acid
(PFOA)) as they are still observed in sewage after years of restrictions. Particulate matter with aerodynamic size ≤10 µm
(PM₁₀) collected above a scaled-down activated sludge tank treating domestic sewage for a population >10,000 people in the
UK were analysed for a range of short-, medium- and long-chain PFAS. Eight PFAS including perfluorobutanoic acid (PFBA),
perfluorobutanesulfonic acid (PFBS), perfluoroheptanoic acid (PFHpA), perfluorohexanesulfonic acid (PFHxS), PFOA,
perfluorononanoic acid (PFNA), PFOS and perfluorodecanoic acid (PFDA) were detected in the PM₁₀. The presence of legacy
PFOA and PFOS in the PM₁₀ samples, despite being restricted for over a decade, raises concerns about their movement through
domestic and industrial sewage cycles. The total PFAS concentrations in PM₁₀ were 15.49 pg m⁻³ and 4.25 pg m⁻³ during
Autumn and Spring campaigns, respectively. PFBA was the most abundant PFAS, suggesting a shift towards short chain PFAS
use. Our results suggest that WWT processes such as activated sludge aeration could aerosolise PFAS into airborne PM.

## 23 1. Introduction

Particulate matter (PM) is a critical component of air pollution and has significant implications for environment (Boucher et
al., 2013; Chen et al., 2021; Taylor and Penner, 1994; Zhang et al., 2023) and human health (Pope III et al., 2020; Vohra et
al., 2021; Zhou et al., 2024). PM with aerodynamic diameter ≤10 µm (PM₁₀) is of particular concern because they are known
to penetrate into the human respiratory system and cause severe health effects (Abbey et al., 1995; Pope III et al., 1992). The
chemical composition of PM is very complex, and it can contain thousands of organic compounds (Goldstein and Galbally,





2007) including persistent organic pollutants (POPs) and new and emerging pollutants (NEPs) such as per- and polyfluoroalkyl
substances (PFAS) (Kourtchev et al., 2022; Zhou et al., 2021; Zhou et al., 2022).
PFAS, commonly referred to as "forever chemicals", are a large group of synthetic organic compounds. PFAS are thermally
and chemically inert due to the strong carbon-fluorine bonds (Buck et al., 2011) and therefore they are widely used in the
production of numerous consumer goods such as water and thermal-resistant apparel, engine oil, cooking wares, etc (Glüge et
al., 2020). PFAS are known for their environmental persistence and bioaccumulation potential (Buck et al., 2011; Lesmeister
et al., 2021). Many PFAS are shown to have negative health effects e.g., endocrine disruption, cancer, and liver disease (Fenton
et al., 2021; Sunderland et al., 2019).
Perfluorooctanoic acid (PFOA) and perfluorooctanesulfonic acid (PFOS) are the most scrutinised PFAS due to their
environmental persistence and human health effects (Beach et al., 2006; Saikat et al., 2013; US EPA, 2024b; Zareitalabad et
al., 2013). In 2009, the Stockholm Convention on POPs included PFOS and its salts in Annex B of restricted compounds.
Further, in 2019 and 2022, PFOA and perfluorohexanesulfonic acid (PFHxS) were added to Annex A of compounds for
elimination. Despite being restricted for more than a decade, these compounds are still observed in various environmental
matrices (Li et al., 2022; Nguyen et al., 2017; Xiao et al., 2015; Zhou et al., 2022). Shortly after the introduction of restrictions
for several PFAS, they were replaced with short-chain and other new-generation PFAS that are thought to be less hazardous
(Brendel et al., 2018; Wang et al., 2013, 2015). These include perfluorobutanesulfonic acid (PFBS), fluorotelomer sulfonates
(FTS), and hexafluoropropylene dimer acid (HFPO-DA, more commonly known as GenX) (Wang et al., 2013, 2015). Recent
studies indicated that numerous replacement PFAS could potentially have similar health effects to those of the legacy ones
(Gomis et al., 2018; Liu et al., 2020; Solan and Lavado, 2022).
The majority of reports on PFAS pollution have predominantly focused on drinking water (Domingo and Nadal, 2019), surface
water (Podder et al., 2021), sewage (Lenka et al., 2021), and soil matrices (Brusseau et al., 2020). Therefore, most of the
current regulations on PFAS are focused on water matrices (Directive (EU) 2020/2184, 2020; US EPA, 2024a). There is
growing evidence that PFAS can transfer from contaminated waters via aerosolisation/volatilisation into atmosphere (Ahrens
et al., 2011; Johansson et al., 2019; Shoeib et al., 2016; Qiao et al., 2024).
Laboratory simulation experiments have shown that the aeration of PFAS-contaminated water leads to formation of aerosolised
PFAS (Nguyen et al., 2024; Pandamkulangara Kizhakkethil et al., 2024). The extent of PFAS aerosolisation has a clear
dependence on the PFAS carbon chain length and functional groups (Johansson et al., 2019; Pandamkulangara Kizhakkethil
et al., 2024; Reth et al., 2011).
Wastewater treatment techniques such as activated sludge (AS) and secondary extended aeration which involve vigorous
aeration/mechanical turbulence, could lead to the aerosolisation/volatilisation of PFAS from contaminated wastewater
effluents (Ahrens et al., 2011; Shoeib et al., 2016). PFAS were detected in gas phase and total suspended particles (TSP) near
the aeration tanks and secondary clarifier in a WWTP in Canada (Vierke et al., 2011). Airborne PFAS were also observed at
WWTPs that employ treatment techniques such as AS, secondary extended aeration, and facultative lagoons in Canada (Shoeib
et al., 2016). PFAS, including restricted PFOS, were identified in the TSP and gas phase above aeration tanks in a WWTP in



northern Germany (Weinberg et al., 2011). A more recent study by Qiao et al. (2024) identified PFAS in both gas and particle
phases above the influent and aeration tanks at a WWTP in China.
Limited studies have assessed the PFAS emission associated with inhalable PM fraction (e.g., $PM_{10}$) during WWT processes.
For example, a recent study identified PFAS in the 11 PM size fractions between 0.1 µm to 18 µm, collected from three
WWTPs in Hong Kong, China (Lin et al., 2022). These WWTPs (largest in Hong Kong) utilised treatment techniques such as
AS and chemically enhanced primary treatment (CEPT) to treat sewage from industrialised areas. The study reported that
atmospheric PFAS in WWTPs (e.g., PFOS, PFOA, PFBS and perfluorobutanoic acid (PFBA)) are primarily distributed in
aerosol particles with aerodynamic diameter ≤10 µm. Additionally, the distribution of PFAS depend on the type of WWT
process, nature of sewage, and aerosol properties (e.g., organic content, presence of microbes, etc.) (Lin et al., 2022). This
suggests that PFAS levels in inhalable PM, and thus the associated health risks, will vary based on the location and the type of
sewage being treated. European countries have restricted the production and use of several PFAS such as PFOS, PFOA,
PFHxS, and C9–C14 perfluorocarboxylic acids (PFCA) (Directive (EU) 2020/2184, 2020;  ECHA, 2022a; ECHA, 2022b).
Nevertheless, the restricted PFOA and PFOS are still observed in wastewater effluents (Eriksson et al., 2017; Gobelius et al.,
2023; Moneta et al., 2023; Müller et al., 2023; Semerád et al., 2020) raising a question whether these chemicals could be
aerosolised during open air aeration WWT processes. To the best of our knowledge, there are no studies assessing the PFAS
levels in $PM_{10}$ at European WWTPs. Furthermore, $PM_{10}$ associated emission of PFAS from WWTPs have been assessed only
for a limited number of PFAS.
As highlighted in the reviews by Phong Vo et al. (2020) and O'Connor et al. (2022), domestic wastewater has been reported
to contain significant levels of PFAS, albeit at concentrations lower than those typically found in industrial effluents. Despite
this, studies on PFAS atmospheric emissions from sewage have primarily focused on WWTPs processing industrial effluents
or a mix of industrial and domestic sources. Consequently, a knowledge gap exists regarding the atmospheric fate of PFAS,
particularly their association with $PM_{10}$ aerosols, during the treatment of domestic wastewater, especially under conditions of
vigorous aeration.
The aim of the current study is to assess the aerosolisation potential of PFAS during WWTP process that involves vigorous
aeration steps. Special emphasis is given to (a) legacy PFAS, such as PFOS and PFOA, as they are still observed in sewage
after 15 and 5 years of restrictions, respectively, and (b) new generation and replacement PFAS such as FTS. To achieve this,
$PM_{10}$ samples collected from a scaled-down AS tank processing domestic wastewater (from a population of > 10,000 people)
in the United Kingdom (UK) were screened for 15 PFAS (C4–C11) including legacy and new-generation replacement
compounds such as FTS.



## 2. Method

### 2.1 Materials and chemicals

The materials and chemicals include: 10 mL headspace glass vials (Chromacol 10-HSV, Thermo Scientific); metal screw caps (Chromacol 18-MSC, Thermo Scientific); polytetrafluoroethylene (PTFE) septa (Chromacol 18-ST101 Thermo Scientific); PTFE membrane filter (Iso-Disc PTFE-13−4, 13 mm × 0.45 μm); glass fiber filters (GFF) (47 mm, Advantec®, Model No. GB-100R); EPA 533 PAR mix containing 25 PFAS i.e., PFBA, PFPeA, PFHxA, PFHpA, PFOA, PFNA, PFDA, PFUdA, PFDoA, HFPO-DA, PFMPA, PFMBA, 3,6-OPFHpA, L-PFBS, L-PFPeS, PFHxS, L-PFHpS, PFOS, 4:2 FTS, 6:2 FTS, 8:2 FTS, NaDONA, 9Cl-PF3ONS, 11Cl-PF3OudS, PFEESA each having a concentration of 0.5 ng μL$^{-1}$ (Wellington laboratories Inc, Canada); EPA533ES isotope dilution standard mixture containing 16 mass labelled ($^{13}$C) PFAS i.e., M3PFBS, M5PFHxA, M6PFDA, M3PFHxS, M8PFOS, MPFBA, M5PFPeA, M4PFHpA, M8PFOA, M9PFNA, M7PFUdA, MPFDoA, M2-4:2 FTS, M2-6:2 FTS, M2-8:2 FTS and M3HFPO with the concentrations of 0.5–2.0 ng μL$^{-1}$; liquid chromatography (LC)-mass spectrometry (MS) grade water (Optima ™, Fisher Scientific); methanol, LC-MS grade (Optima ™, Fisher Scientific); formic acid, LC-MS grade (Optima™, Fisher Chemicals); ammonium acetate, LC-MS grade (Optima™, Fisher chemicals). The full names of the listed chemicals are given in the Table S1 and S2 of supplement.

### 2.2 Sampling site

The PM$_{10}$ samples were collected above a scaled-down AS tank processing municipal wastewater equivalent to that of a population > 10,000 people (the location of the facility is anonymised due to a non-disclosure agreement). The AS tank, constructed from high-density polyethylene (HDPE), contains an aeration basin of volume 3.06 m$^3$. The aeration basin of the AS tank is connected to a secondary clarifier (of volume 0.86 m$^3$) where sewage, after aeration, is allowed to settle. The AS tank continuously receives and processes primary treated sewage (with a solid retention time (SRT) of 10 days) from the parent large-scale WWTP using pumps.

### 2.3 PM$_{10}$ sample collection

The MiniVol™ tactical air sampler (Air Metrics, United States of America) used for PM$_{10}$ sampling was installed near the aeration tank with the sampling head slightly above the rim of the tank. The PM$_{10}$ samples were collected on GFF at 10 L min$^{-1}$. Prior to sampling, GFF were baked at 450 °C for 24 h to eliminate potential organic contaminants. The samples were collected during two sampling periods: between (1) 2 October 2023–6 October 2023 and (2) 4 March 2024–8 March 2024. PM$_{10}$ samples were collected separately during day and night. Sampling dates and duration are given in Table 1.

GFF with PM$_{10}$ were rolled using prewashed stainless-steel tweezers, keeping aerosol content inside, and placed into a prewashed 10 mL headspace glass vial. 5 mL methanol (LC-MS grade) was added to the vial to disinfect the filters from potential pathogenic microorganisms and extract the organic compounds including PFAS. The samples were then stored at 5



°C until the day of analysis. The vials, PTFE septa, and metal screw caps were prewashed with LC-MS grade methanol and
dried before use to remove potential PFAS contamination. PFAS leaching from the vials, PTFE septa, and metal screw caps
was assessed in another study which reported minimal PFAS leachables from these consumables (Kourtchev et al., 2022).
Various types of blanks were prepared to evaluate possible PFAS contamination from handling the filters. These include: 1)
baked filters and 2) baked filters placed in MiniVol® air sampler and collecting air above the AS tank at 10 L min$^{-1}$ for 2 min.
It is important to note that the use of GFF and quartz fiber filters (QFF) during PM sampling has been reported to cause positive
sampling artefacts, such as the adsorption of gas-phase organic compounds (Turpin et al., 1994). Previous studies have shown
that certain PFAS, such as PFOS and PFOA, can transition from aqueous aerosols to the gas phase (Ahrens et al., 2012;
McMurdo et al., 2008). As a result, the GFF used in our study may also include a small fraction of gas-phase PFAS.
Consequently, the reported PM$_{10}$ concentrations of PFAS in our study might be slightly overestimated.

**Table 1** PM$_{10}$ sample collection dates and duration

| Sampling date | Sample type | Sampling duration (h) |
|---|---|---|
| 2 October 2023 | Day sample | 3.3 |
| | Night sample | 9.2 |
| 3 October 2023 | Day sample | 5.7 |
| | Night sample | 17.9 |
| 4 October 2023 | Day sample | 5.7 |
| | Night sample | 18 |
| 5 October 2023 | Day sample | 5.7 |
| | Night sample | 18 |
| 4 March 2024 | Day sample | 1.4 |
| | Night sample | 19.1 |
| 5 March 2024 | Day sample | 4.5 |
| | Night sample | 18.8 |
| 6 March 2024 | Day sample | 4.8 |
| | Night sample | 18.5 |
| 7 March 2024 | Day sample | 4 |
| | Night sample | 19 |





## 2.4 Extraction and analysis of PM$_{10}$ GFF samples

The GFF samples stored in methanol were spiked with internal standards (IS), a mixture of 16 $^{13}$C PFAS labelled compounds
(EPA533 ES, Wellington laboratories Inc, Canada) at concentrations 20 ng L$^{-1}$ for M2-4:2 FTS, M2-6:2 FTS, and M2-8:2FTS
and 5 ng L$^{-1}$ for the remaining compounds and extracted using the procedure published in Kourtchev et al. (2022).
Briefly, the vial content was subjected to ultrasonic agitation for 40 min. The methanol extracts were then filtered through a
prewashed 0.45 µm PTFE syringe filter. The PTFE filters used in our study were assessed for PFAS leaching potential in
Kourtchev et al. (2022). Minimal leaching of PFAS was observed from the PTFE filters after purging them with 5 mL LC-MS
grade methanol three times (total volume of 15 mL) (Kourtchev et al., 2022). The extracts were then reduced by volume to 1
mL under gentle nitrogen flow.
The methanolic extract was then topped up with 4 mL of LC-MS grade water providing the 80:20 (v/v) water: methanol ratio
required for the online solid phase extraction (SPE) (Kourtchev et al., 2022). The vial content was homogenised by vortex
mixing and analysed using online SPE LC-high resolution mass spectrometry (HRMS) using the method published elsewhere
(Kourtchev et al., 2022).
The online SPE and chromatographic separation was carried out using EQuan MAX Plus Thermo Scientific™ Vanquish™
UHPLC system using a Thermo Scientific™ TriPlus™ RSH autosampler. Online SPE was performed using a Thermo
Scientific™ Hypersil GOLD aQ Column, 20 × 2.1 mm, 12 µm column. 0.1 % formic acid in water was used as the loading
phase for the online SPE. Following online SPE, the chromatographic separation was achieved using Waters® CORTECS
C18 Column, 90 Å, 100 × 2.1 mm, 2.7 µm analytical column. The eluents used for chromatographic separation were A) 2 mM
ammonium acetate in 10 % methanol and B) 100 % methanol. A Q Exactive™ Focus Hybrid Quadrupole-Orbitrap™ Mass
Spectrometer (Thermo Fisher, Bremen, Germany) fitted with electrospray ionisation (ESI) (Ion Max™) source was employed
for the mass spectrometric analysis. The mass spectrometric analysis was performed in single ion monitoring (SIM) negative
ionisation mode. The mass spectrometer was calibrated prior to analysis to have a mass accuracy of ≤ 5 ppm. The limit of
detection (LOD) values for the analytes in this study were similar to those reported by Kourtchev et al. (2022), with the
exception for PFBA. The LOD for PFBA was 1.47 ng L$^{-1}$, which is higher than the value reported by Kourtchev et al. (2022)
and could potentially be due to higher background levels of the analyte in the system blanks.

## 2.5 Quality assurance (QA) and quality control (QC)

Several steps were taken to ensure the QA and QC during the sampling and analysis. PFAS-containing consumables were
avoided as much as possible during the sampling, extraction, and LC-HRMS analysis. To prevent accumulation of PFAS in
the LC-HRMS system, prior to the analysis, the system was flushed with the mobile at composition of 60:40 A: B (A: 2 mM
ammonium acetate in 10 % methanol and B: 100 % methanol) and 0.3 mL min$^{-1}$ flow rate, overnight (Kourtchev et al., 2022).
System suitability tests (SST) were performed before the analysis of each batch to ensure the adequate performance of the LC-
HRMS system. Pass criteria were evaluated based on chromatographic peak area and height, retention times, mass accuracy,





and peak tailing factors. System blanks ("zero volume") and 80:20 water: methanol (v/v) blanks were injected at the start of
the batch, in between the samples, and at the end of the batch to monitor a potential PFAS carry over. The zero volume blanks
and 80:20 water: methanol blanks reported PFAS concentrations less than the method LOD values.
**3. Results and discussion**
**3.1 PFAS composition of $PM_{10}$ above the AS tank**
Figure 1 shows the concentrations of PFAS detected in $PM_{10}$ samples collected above the AS tank during the two sampling
periods. Out of the 15 target PFAS, eight compounds were detected across the collected samples.







**Figure 1** Concentrations of PFBA, PFBS, PFHpA, PFHxS, PFOA, PFOS, PFNA, and PFDA in the $PM_{10}$ samples collected
from the AS tank in October 2023 and March 2024. The absence of data points on certain sampling days indicates that the
compound was either not detected or below the method LOD. The error bars represent the standard deviation of the value from
three replicate injections. The data of Figure 1 are shown in Tables S3 and S4.

The detected PFAS include short-chain PFBA and PFBS, medium-chain PFHpA, PFHxS, PFOA, PFNA, and PFOS, and long-
chain PFDA. The most abundant PFAS detected in the $PM_{10}$ from both sampling campaigns was a short-chain PFBA with a
maximum concentration of $19.6\pm0.8$ pg m$^{-3}$ in October 2023 and $8.8\pm0.9$ pg m$^{-3}$ in March 2024. The concentration of PFAS
detected in the samples from October 2023 followed the order PFBA>PFOS>PFOA>PFDA, with maximum concentrations
recorded at $17.4\pm0.2$ pg m$^{-3}$ for PFOS, $8.1\pm0.4$ pg m$^{-3}$ for PFOA, and $3.7\pm0.1$ pg m$^{-3}$ for PFDA. The samples collected during
March 2024 showed a different pattern, with PFOA ($1.70\pm0.01$ pg m$^{-3}$) having the highest concentration after PFBA, followed
by PFOS at $0.76\pm0.02$ pg m$^{-3}$. It has been reported that aerosolisation of PFAS from contaminated water depends on carbon
chain length and functional groups, with higher aerosol enrichment for long chain PFAS and perfluorosulfonic acids (PFSA)
compared to PFCA (Johansson et al., 2019; Pandamkulangara Kizhakkethil et al., 2024; Reth et al., 2011). However, it is
interesting to note that the PFAS levels in the $PM_{10}$ in our study followed a reverse order with short chain PFBA detected at
higher values.
The detected PFAS have been associated with different sources. For example, PFOS, PFNA, and PFOA have historically been
produced and used in the manufacturing of numerous products, such as firefighting foam, fluoropolymers, textiles, leather,
paper, and lubricants (ATSDR, 2015; Buck et al., 2011; de Alba-Gonzalez et al., 2024; Wang et al., 2014). PFHxS and its
salts/related compounds have been used in applications such as firefighting foam, coatings, electronics and semiconductors,
and polishing agents (in many of these applications PFHxS has been introduced as a replacement for PFOS)
(UNEP/POPS/POPRC.15/7/Add.1, 2019). PFBA and PFBS, have been used as replacements for legacy and longer chain PFAS
(Ateia et al., 2019; Christian, 2024; Wang et al., 2013). PFBA is used in the manufacturing of food packaging materials,
carpets, and fluorosurfactants (Christian 2024; US EPA, 2022). PFBS and PFBS based compounds are used in applications
such as metal plating, as flame retardant, and surfactant (Wang et al., 2013). PFDA, a long chain PFAS identified in the PM
in this study, have been reported as a breakdown product of stain- and grease-proof coatings on food packaging, furniture, and
carpets (Christian, 2024).
Clear differences were observed in the concentrations of PFAS in $PM_{10}$ samples from the two sampling campaigns. In general,
concentrations of all detected PFAS except PFHxS were higher in the samples collected in October 2023 compared to the
March 2024 samples. For example, highest concentration of PFBA reported during the March 2024 period was less than half
of that reported in the October 2023 period. PFNA and PFDA were absent in the samples from March 2024, but they were
detected in the October 2023 samples. The concentrations of PFHxS and PFHpA reported during both sampling periods were
higher than the method LOD but slightly lower than the method limit of quantification (LOQ) values. There are several
potential reasons for the observed seasonal differences in the concentrations of PFAS which include the pH value, density, and



composition of the wastewater. The pH value of the contaminated water has been found to affect the water to atmospheric
transfer behaviour of PFAS (Ahrens et al., 2012; Barton et al., 2007; Pandamkulanagra Kizhakkethil et al., 2024; Vierke et al.,
2013). For example, the average pH of the wastewater in October 2023 was 7.5, whereas the average pH was 9.3 during the
March 2024 sampling campaign. Additionally, the sewage density and potentially the composition were different during the
two sampling periods (the pH and density data are not shown in the manuscript due to the non-disclosure agreement). PFAS
are well known for their sorption to biosolids in sewage (Ebrahimi et al., 2021; Link et al., 2024). During the March 2024
sampling period, the sewage was thicker compared to October 2023, potentially leading to higher sorption of PFAS in the
biosolids and consequently lesser $PM_{10}$ associated emissions. It should be noted that the sewage composition was not static
during the sampling periods. The SRT of the AS tank was 10 days, and the chamber received and processed primary treated
wastewater from the parent WWTP continuously. Therefore, the variation in the sewage composition could potentially explain
the differences in the airborne PFAS concentration Moreover, the surface runoff, linked to rainfall, could also be a factor
influencing the overall PFAS levels, as it may introduce additional contaminants to the wastewater system.
Since the sampling campaigns were conducted at two different seasons, the atmospheric conditions e.g., temperature, relative
humidity (See Fig. S1–S4 of a supplement for the average temperature and relative humidity at the sampling periods) could
also influence the PFAS partitioning to aerosols from the contaminated water (Ahrens et al., 2012). It should be noted that the
absence of PFNA and PFDA in the March 2024 samples could be attributed to lower concentration of these analytes in the
sewage resulting in PFAS $PM_{10}$ bound concentrations below the method LOD.
Several PFAS exhibited day and night variations in $PM_{10}$ samples. For example, the PFBA concentration was higher during
the day compared to the night in specific sampling days of October 2023. On the other hand, PFBA concentration during the
day was close to the background levels during the March 2024 campaign. PFHpA and PFHxS were not detected in the day
samples during both sampling campaigns. Legacy PFOS and PFOA showed higher concentrations during the day on specific
sampling days. The difference in the diurnal concentrations could potentially be due to variability in the composition of the
wastewater at the respective sampling time. The diurnal variations in the environmental conditions such as temperature and
relative humidity could also contribute to the observed higher PFAS concentrations observed in the specific day samples
compared to the night samples.
**3.2 Comparison to previous studies**
The observation of high levels of PFBA in our study is consistent with the results of Weinberg et al. (2011), who identified
PFBA (up to 8.4 pg m$^{-3}$) as the most abundant PFAS in the PM samples (TSP) collected above the aeration tanks of two
WWTPs processing a mixture of domestic and industrial wastewater in Northern Germany. PFBA was also identified as the
dominant ionic PFAS in the atmosphere of WWTPs in other studies (Shoeib et al., 2016; Lin et al., 2022). For example, air
samples collected using sorbent-impregnated polyurethane foam (SIP) passive air samplers at WWTPs employing AS
(processing mixed wastewater), secondary extended aeration (one processing domestic and the other two processing mixed



wastewater), and facultative seasonal discharge lagoons (processing domestic wastewater) in Canada detected PFBA up to
60±21 % of the total PFCA detected (Shoeib et al. 2016). Similarly, Lin et al. (2022) reported PFBA at considerable levels in
the atmosphere near the aeration tanks of two WWTPs and above a WWTP using CEPT (processing wastewater from urban
areas) in Hong Kong, China. The concentrations of PFBA in TSP reported by Lin et al. (2022), with maximum values of 9.17
pg m$^{-3}$ and 15.6 pg m$^{-3}$ near the aeration tanks, which are comparable to the values reported in our study.
The high PM$_{10}$-associated concentration of PFBA in our study could potentially be explained by the recent increase in the use
of short-chain PFAS as a replacement for legacy PFOS and PFOA (Ateia et al., 2019; Wang et al., 2013). PFBA is one of the
most volatile PFAS observed in our study. Further, short chain PFAS such as PFBA and PFBS are reported to have lower
aerosolisation tendencies compared to long chain compounds (e.g., PFOS and PFOA) (Johansson et al., 2019;
Pandamkulangara Kizhakkethil et al., 2024). Despite being volatile and having low aerosolisation tendency, the presence of
PFBA in the PM$_{10}$ aerosols at considerable concentrations in our study could potentially be due to the presence of high levels
of PFBA in the sewage during the sampling period.
The concentrations of PFAS in PM$_{10}$ reported in our study, except for PFHxS, were higher than those estimated by Weinberg
et al. (2011) in the particulate phase (TSP) above the aeration tanks of a WWTP that processed a mix of domestic and industrial
waste in Northern Germany. For example, during the October 2023 sampling period, legacy PFAS such as PFOS and PFOA
were detected in our study at levels up to 17.4 ± 0.2 pg m$^{-3}$ and 8.1 ± 0.4 pg m$^{-3}$, respectively. In contrast, the maximum
concentrations of PFOS and PFOA during March 2024 were 0.76 ± 0.02 pg m$^{-3}$ and 1.70 ± 0.01 pg m$^{-3}$, respectively. Weinberg
et al. (2011) estimated PFOS and PFOA concentrations in the TSP to be up to 0.9 pg m$^{-3}$ and 1.3 pg m$^{-3}$, respectively. The
difference in the PFAS emission levels could be potentially due to the difference in PFAS composition in the wastewater.
PFAS composition in wastewater across European Union (EU) have been reported to differ depending on the region (Lenka
et al., 2021)
The PFDA concentrations of (up to 1.31 pg m$^{-3}$) in the TSP samples reported by Lin et al. (2022) above the aeration tanks of
WWTPs in Hong Kong, China were lower than the PFDA levels observed in our study during the October 2023 period (3.7 ±
0.1 pg m$^{-3}$). However, for other PFAS compounds such as PFBS, PFHxS, PFHpA, PFOA, PFOS, and PFNA, Lin et al. (2022)
reported considerably higher values in the TSP samples than those observed in our study. Lin et al. (2022) investigated the
distribution of PFAS across 11 PM size fractions (ranging from 0.1 μm to >18 μm) collected from three WWTPs (two using
aeration and one using CEPT), as well as a landfill and two reference sites. PFOS in PM from all studied WWTPs (treating
urban wastewater) showed major distribution around the PM fractions with aerodynamic size between 0.1 and 10 μm.
Similarly, PFBA and PFBS were also found to be primarily associated with particles of aerodynamic size less than 10 μm,
indicating that the PM$_{10}$ collected in our study could have potentially captured a majority of the PFAS bound particles. The
reported values in our study therefore provide insights into the total aerosol bound emissions of studied PFAS during the WWT
process.
The PFAS reported in our study were significantly lower than the PM (TSP) values reported by Vierke et al. (2011) (processing
wastewater from Ontario, an urban area in Canada). For example, the average PM concentrations of PFOS and PFOA above



the aeration tank of a WWTP in Canada study were 3900 pg m$^{-3}$ and 71 pg m$^{-3}$, respectively (Vierke et al., 2011). Similarly,
Qiao et al. (2024) also reported considerably higher values for legacy PFOS (1.7–65.1 pg m$^{-3}$) and PFOA (3.1–101 pg m$^{-3}$) in
the TSP samples above the influent and aeration tanks of two WWTPs (one processing domestic wastewater and the other one
processing industrial wastewater) in China.
It is interesting to note that the PFAS levels in PM$_{10}$ reported in our study are comparable to those reported by Weinberg et al.
(2011) in the TSP samples, which is the only study that investigated atmospheric PFAS levels in European WWTPs. The
similarity in the TSP and PM$_{10}$ concentrations could be due to PFAS being associated mainly with aerosols having aerodynamic
size less than 10 µm as shown for several type of sewage in Lin et al. (2022). In contrast, higher PFAS levels in TSP samples
were reported in all other studies conducted at WWTPs in Canada and China (Lin et al., 2022; Vierke et al., 2011; Qiao et al.,
2024). The differences of PFAS levels in PM could potentially be due to variations in wastewater composition in these regions.
For example, the WWTP studied by Vierke et al. (2011) is situated in Ontario, a heavily industrialised city in Canada. Similarly,
the WWTPs investigated by Lin et al. (2022) and Qiao et al. (2024) are located in China, one of the most heavily industrialised
countries in the world.  The facility in our study processes sewage mainly from households (for approximately 30,000 people)
rather than industries, which may contain lower PFAS levels in the sewage and thus in aerosol. The total PFAS concentrations
associated with PM$_{10}$ fractions in our study were 15.49 pg m$^{-3}$ in October 2023 and 4.25 pg m$^{-3}$ in March 2024 (see Table S3
and S4 of supplement), which is comparable (2-13 pg m$^{-3}$) to that in the TSP from mixed wastewater in Northern Germany
reported by Weinberg et al. (2011).

## 291    4. Conclusion

In this study, we investigated, for the first time, the PFAS concentrations associated with the health-relevant PM$_{10}$ fraction of
airborne aerosols emitted during the AS aeration process at a WWTP processing domestic wastewater. PM$_{10}$ samples were
collected over two sampling campaigns at two different seasons (i.e., October 2023 and March 2024) above a scaled-down AS
tank consisting of an aeration basin of volume ~3 m$^3$, treating wastewater equivalent to > 10,000 people. Eight PFAS were
observed across the collected PM$_{10}$ samples. These include legacy PFOS and PFOA, which were detected up to concentrations
of 17.4±0.2 pg m$^{-3}$ and 8.1±0.4 pg m$^{-3}$, respectively in the samples from October 2023.
The presence of legacy PFOS and PFOA in the PM even after a decade-long restriction raises concern and suggests that PFOS
and PFOA-containing products are still in use or in the recirculation cycle. More studies are needed to understand if these
legacy compounds could have been formed in the wastewater during the treatment process from the degradation of precursor
compounds such as fluorotelomer alcohols (FTOH) perfluorooctane sulfonamides (FOSA), perfluorooctane
sulfonamidoethanols (FOSE) as suggested by Dauchy et al. (2017) and Xiao (2022).
Presence of PFBA at high concentrations in the collected samples potentially suggests the increased shift towards the use of
short-chain PFAS as a replacement for legacy PFAS.



Our results indicate that WWT processes involving aeration could aerosolise and transfer PFAS into the atmosphere.
Considering the sheer number of different PFAS that are in the production and used today, the estimated total PFAS
concentrations could potentially represent only a fraction of the actual emissions during the aeration process.
To the best knowledge, this is the first study to investigate the presence of PFAS in the $PM_{10}$ fraction of the airborne aerosols
from the AS aeration process in a WWTP in the UK and Europe.
**Data availability**
The data are not publicly accessible due to a non-disclosure agreement with a wastewater treatment company, which is also
anonymised. The data of Figure 1 are shown in Tables S3 and S4.
**Competing interests**
Some authors are members of the editorial board of journal ACP. The authors have no other competing interests to declare.
**Acknowledgements**
The authors wish to acknowledge Coventry University QR funding for providing Trailblazers PhD studentship (acquired by
Dr Ivan Kourtchev) to Jishnu Pandamkulangara Kizhakkethil and the Centre for Agroecology, Water and Resilience (CAWR),
Coventry University for providing financial support.
**Author contributions**
JPK conceived the study and performed filed measurements, sampling, sample analysis, data processing and interpretation.
AB supervised the project. ZS supervised the project and provided resources. IK conceived and led the study, supervised the
project, obtained funding, provided the resources, performed field measurements, sampling, data interpretation. JPK and IK
prepared the original draft of the paper. All authors contributed to reviewing and editing the manuscript.

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
