# Peer review of "Measurement report: Per- and polyfluoroalkyl substances (PFAS) in particulate matter $(PM_{10})$ from activated sludge aeration"

_EGUsphere, 2024_

## Author Comment (AC1)

Responses to the **Reviewer 2** comments

**Comment:** In this article, the authors measure the concentrations of 15 legacy and emerging PFAS in PM10 collected from a scaled down activated sludge aeration tank. This study was conducted at two different time points, Oct 2023 and March 2024 using domestic sludge from a wastewater treatment plant in the UK. This manuscript is well written and fills a knowledge gap regarding PFAS aerosolization from domestic sludge wastewater treatment processes. I recommend this paper for publication following revision.

**Response:** We would like to sincerely thank Reviewer 2 for recognising importance of our work in bridging the knowledge gap on the aerosolisation of PFAS from domestic wastewater treatment processes. We also appreciate his/her recommendation of our work for publication following the revision, as well as his/her insightful comments that have helped strengthen our work. Our detailed responses to the reviewer's comments are provided below in blue. The new text added to the revised manuscript is shown below in italics.

**General Comments:**

I strongly recommend including both the LOD and LOQ values in the SI. I also recommend reporting the lab, field, and system blanks as well.

**Response:** We thank the reviewer for this comment. The LOD, LOQ, and values for different types of blanks (i.e., filter blanks, field blanks, and system blanks) have been now reported in SI in tables S5-S7.

In line 90, the authors mention screening from 15 PFAS but then in line 97 state that the EPA 533 PAR mix, which contains 25 PFAS, was used. Please list which PFAS were targeted and provide justification for why some were targeted, and others were not.

**Response:** The analysis of the $PM_{10}$ samples was done using the analytical method published elsewhere (Kourtchev et al., 2022). This analytical method is fully validated for analysis of 15 out of the 25 compounds present in the EPA 533 PAR mixture. To ensure the accuracy, reliability, and reproducibility of analytical results we only focused on those fully validated analytes. This has now been addressed in Section 2.4 of the main text: "*The analytical method is validated for screening and quantifying 15 PFAS including PFBA, PFPeA, PFBS, 4:2 FTS, PFHxA, PFPeS, PFHxS, PFHpA, PFOA, PFHpS, PFNA, PFOS, 8:2 FTS, PFDA, and PFUdA. Therefore, to ensure the accuracy, reliability, and reproducibility of analytical results, the current study focused only on those fully validated analytes.*" (lines 148-151, page 6 of the revised main text).

Please provide further clarity regarding the scaled-down AS tank as well as the sample collection. How much smaller was the scaled-down tank compared to the large-scale WWTP? How far from the tank and how high above the rim of the tank was the MiniVol placed?

**Response:** Thank you for this comment. Due to our strict non-disclosure agreement (NDA) with the wastewater treatment company, we cannot provide additional details on the scale-down AS tank. However, we strongly believe that even without this information, our results still provide critical information on PFAS emissions, particularly for legacy PFAS that remain in wastewater streams despite their ban, highlighting their persistence in sewage processing and the challenges of their removal. Moreover, our results highlight their distribution beyond expected environments, including partitioning between water and aerosol phases in sewage processing.

The position of the air sampler relative to the AS tank has been now included in Section 2.3 of the main text: "*The MiniVol™ tactical air sampler (Air Metrics, United States of America) used for $PM_{10}$ sampling was installed near the aeration tank (<0.2 m from the aeration tank) with the sampling head slightly above the rim of the tank (10 cm above)*." (lines 114-115, page 4 of the revised main text).

Why is there such a difference in the sampling time for day and night samples? Would the short day sampling periods (1.4 h to 5.7 h) account for the low detection frequency for day samples, especially as compared to the night?

**Response:** We thank the reviewer for this important comment. The day period sampling started after 10.00 AM and finished before 3.00 PM while the night period sampling started after 3.00 PM and finished before 10.00 AM the next morning. This is because the working time of the staff in the facility is between 9.30 AM and 3.30 PM. Moreover, the MiniVol™ tactical air sampler does not allow automatic filter change requiring us to manually replace filters within the facility's access hours. Due to these restrictions, the day samples are lower in sampling duration than the night time samples. As the reviewer pointed out, the low detection frequency of PFAS in the day samples could be indeed attributed to lower aerosol mass load on the filters caused by the shorter sampling duration. This has been now included in Section 3.1 of the main text: "*The shorter sampling duration of day samples compared to the night samples likely led to a lower aerosol mass load on the filters, resulting in several PFAS mass loads below the LODs, which could explain the observed diurnal differences in PFAS concentrations.*" (lines 248-250, page 10 and 11 of the revised main text).

Line 135: Were the filters spiked with IS and then the 5mL of methanol added? Or were the filters sitting in methanol and then, when ready for extraction, the solution of methanol with a filter was spiked with IS?

**Response:** After sampling, the filters were stored in 10 mL headspace vials and 5 mL of methanol was added immediately to decontaminate the filters from potentially pathogenic microorganisms present in sewage. The samples were then stored at 5 °C until the day of extraction. At the time of extraction, the internal standard was spiked into the vial containing the filter soaked in 5 mL of methanol. This was discussed in lines 119-122 of Section 2.3 in the main text of the original submission.

Section 2.5 – What were the recoveries/extraction efficiencies for the targeted PFAS? Please report these in the SI.

**Response:** Thank you for this comment. The extraction efficiencies of the targeted PFAS have now been included in SI as Table S8.

Line 225: This section seems incomplete and highly speculative. PFBA was detected during the day, twice, in the October sampling period and only once was it higher than the night concentration. I don't know that it can be claimed that the differences are attributable to diurnal variations when the sampling periods for day and night are so different. For PFHpA and PFHxS, the measured concentrations during the night sampling are so low (how close to the LOD/LOQ?) that it might simply be that the sampling time during the day was not long enough to collect sufficient mass to be quantified.

**Response:** We thank the reviewer for this comment. We agree that PFHpA and PFHxS were not detected in the day samples potentially due to the low mass load in the day time samples associated with the shorter sampling duration. PFHpA and PFHxS were detected above LOD but below LOQ values in the nighttime samples. However, in the case of PFBA, the concentrations detected in the

samples were higher than the LOD/LOQ values. This suggests that the differences in the PFBA concentrations during the day and night reported in our study could potentially be attributed to the diurnal variations in the environmental conditions. To clarify this, we have now added the following text: "*The shorter sampling duration of day samples compared to the night samples likely led to a lower aerosol mass load on the filters, resulting in several PFAS mass loads below the LODs, which could explain the observed diurnal differences in PFAS concentrations.*" (line 248-250, page 10 and 11 of the revised main text).

I recommend adding a limitations section or at least paragraph on the limitations of the study. It may be beneficial to include the following: wastewater was not analyzed for PFAS, targeted PFAS was limited to 15 out of thousands, and collection of both gas- and particle-phase PFAS. Studies (see Ao et al., 2024 - 10.1016/j.jhazmat.2023.133018) have also detected polyfluoroalkyl phosphate esters (PAPs) at high concentrations in household dust as well as in food-contact materials, cosmetics, and other consumer products. They've also be shown to biodegrade into PFOA (8:2 diPAP) and to other PFCAs (see Lee et al., 2010 - https://doi.org/10.1021/es9028183 and Liu and Liu 2016 - 10.1016/j.envpol.2016.01.069).

**Response:** A limitation section has been now included in the main text: "*Future research should consider simultaneous characterisation of wastewater PFAS levels alongside PM measurements to improve understanding of the relationship between airborne PFAS emissions. Expanding the range of monitored PFAS beyond the 15 fully validated targets in our study, particularly including neutral PFAS such as FTOHs and FOSEs, would enhance understanding their role in the WWTP aerosolisation. Additionally, incorporating gas-phase sampling would be valuable in assessing the potential partitioning of PFAS into the gaseous phase, further refining our understanding of their atmospheric behaviour.*" (lines 326-331 page 13 of the revised main text).

Additionally, while the authors note that anonymity of the WWTP limits the environmental data they can share, can the authors provide any comment on possible nearby sources of ambient PFAS (e.g., other fluorochemical manufacturing plants or point sources). Wind direction and wind speed data could have also been informative.

**Response:** We thank the reviewer for recognising the challenges caused by the NDA in revealing the environmental data. As detailed in the manuscript, the major source of sewage to the WWTP is from households. Unfortunately, we are unable to share the wind direction/wind speed data as this would require revealing the WWTP location, which is restricted by the NDA. As mentioned above, even without this information our results provide critical insights into PFAS, particularly for legacy PFOA and PFOS, emissions from the process despite their ban, highlighting their persistence, the challenges of their removal, and their distribution beyond water into other environmental media.

The authors may also find it useful to expand their discussion to include the implications of detecting PFAS in PM10 from aerosolized domestic waste. What does this mean for long-range transport and human exposure? As more stringent regulations are placed on emissions of PFAS from major fluorochemical plants, domestic-related emissions are likely to become more important.

**Response:** We appreciate the reviewer's suggestion. Due to NDA restrictions, we are limited in how much we can elaborate on this aspect. The potential contribution of WWTPs to PFAS pollution remains an open question that warrants further investigation. While we cannot draw definitive conclusions at this stage, we have acknowledged this limitation in the manuscript. It must be noted that for the limitations outlined above, the article was submitted as a Measurement Report rather

than a full research article. The following text has been added to the manuscript: "*Future research should consider simultaneous characterisation of wastewater PFAS levels alongside PM measurements to improve understanding of the relationship between airborne PFAS emissions. Expanding the range of monitored PFAS beyond the 15 fully validated targets in our study, particularly including neutral PFAS such as FTOHs and FOSEs, would enhance understanding their role in the WWTP aerosolisation. Additionally, incorporating gas-phase sampling would be valuable in assessing the potential partitioning of PFAS into the gaseous phase, further refining our understanding of their atmospheric behaviour.*" (lines 326-331 page 13 of the revised main text).

Specific Comments:

Line 35: Specify types of cancer associated with exposure to PFAS.

**Response:** The examples of the type of cancer associated with PFAS exposure have been included in Section 1 of the main text: "Several PFAS are shown to have negative health effects e.g., endocrine disruption, cancer including *kidney and testicular cancer,* and liver disease (Fenton et al., 2021; Sunderland et al., 2019)" (line 35-36, page 2 of the revised main text).

Lines 185 – 187: Does the composition of the contaminated water (particularly the organic content of wastewater) also influence the degree of aerosolization?

**Response:** We thank the reviewer for this comment. A recent study on PFAS sea spray aerosol (SSA) simulation experiments by Sha et al., (2024) reported enhanced PFAS enrichment in the generated SSA when organic matter was introduced to artificial sea water. However, it should be noted that the aerosol generation mechanism from WWTPs could be different from that of SSA. This information have been now updated in Section 3.1 of the main text: "It has been reported that aerosolisation of PFAS from contaminated water depends on carbon chain length, functional groups *and organic content,* with higher aerosol enrichment for long chain PFAS and perfluorosulfonic acids (PFSA) compared to PFCA (Johansson et al., 2019; Pandamkulangara Kizhakkethil et al., 2024; Reth et al., 2011; *Sha et al., 2024*)." (lines 197-200, page 9 of the revised main text).

Line 201: I think it's interesting that PFOA was detected in all day samples during the October period but not in the March samples. The authors comment on differences in sewage composition affecting the measured concentrations in PM10, but what about domestic activities that occur in the Fall vs the Spring that may contribute to this? Are the authors able to provide insight into this seasonal difference beyond the sewage differences? I recognize this is probably quite difficult as there are many different sources of PFAS, but this line of thinking could yield interesting theories and questions. However, it may be that the authors simply state (if they agree) that this suggests that there are seasonal variations in household activities that may affect sewage concentrations.

**Response:** We agree with the reviewer that differences in household activities during different seasons could potentially influence the sewage concentrations of the PFAS and thereby the PFAS levels in the PM. This has now been included in Section 3.1 of the main text: "*Seasonal variations in PFAS $PM_{10}$ levels could also be due to changes in household activities throughout the year and thus concentrations in domestic wastewater entering the WWTP.*" (lines 234-235, page 10 of the revised main text).

Line 249: Is it possible that PFBA and PFBS, which are volatile, are present in the gas-phase and sorbed to the GFF? Or is this unlikely. Can the authors provide insight into this?

**Response:** PFBA has a vapor pressure of 2.92 mm Hg at 25°C (Zhang and Suuberg, 2023), whereas PFBS has a vapor pressure of 0.0268 mm Hg at 25°C (Pubchem, 2025) suggesting that they are semi-volatile.

Nevertheless, as the reviewer pointed out, a fraction of PFBA or PFBS volatilised during the WWTP process could be adsorbed onto the glass fiber filters (GFF). This was acknowledged in the initial submission i.e. "It is important to note that the use of GFF and quartz fiber filters (QFF) during PM sampling has been reported to cause positive sampling artefacts, such as the adsorption of gas-phase organic compounds (Turpin et al., 1994). Previous studies have shown that certain PFAS, such as PFOS and PFOA, can transition from aqueous aerosols to the gas phase (Ahrens et al., 2012; McMurdo et al., 2008). As a result, the GFF used in our study may also include a small fraction of gas-phase PFAS. Consequently, the reported $PM_{10}$ concentrations of PFAS in our study might be slightly overestimated." (lines 127-131, section 2.3).

Line 252 and Line 257: Are the values estimated or measured by Weinberg et al. (2011)?

**Response:** The values reported by Weinberg et al. (2011) are measured values, not the estimated ones. This word 'estimated' is now replaced with '*measured*' in Section 3.2 of the main text.

Line 285: Specify the two cities where Lin et al., (2022) and Qiao et al. (2024) sampled. It seems a bit like apples and oranges to specify Ontario, Canada and then all of China.

**Response:** This has been now addressed in Section 3.2 of the main text: "Similarly, the WWTPs investigated by Lin et al. (2022) and Qiao et al. (2024) are located in China (*Hong Kong and Tianjin, respectively*), one of the most heavily industrialised countries in the world." (lines 298-299, page 12 of the revised main text).

Line 287 – 290: Is this a useful comparison? Are the two studies comparing the same number and types of PFAS? Perhaps it would be more comparable to sum report the total PFAS concentrations for only the matched PFAS. Also, are the LOD/LOQs for this study and Weinberg et al. (2011) comparable?

**Response:** The authors thank the reviewer for this comment. The ionic PFAS in TSP detected by Weinberg et al. (2011) include PFBS, PFHxS, PFBA, PFPeA, PFHxA, PFHpA, PFOA, PFNA, PFDA, PFUdA, PFDoA, and perfluorosulfonamide (PFOSA). Our method is able to detect and thus quantify all the ionic PFAS (except PFDoA and PFOSA) detected by Weinberg et al., (2011). By removing the concentrations reported for PFDoA and PFOSA, the ∑PFAS concentrations in TSP are in the range of 1-10.8 pg m$^{-3}$, which is comparable with already reported 2-13 pg m$^{-3}$. The LOD and LOQ values reported by Weinberg et al. (2011) are higher than our LOD and LOQ values. For example, the LOD and LOQ for PFOS reported by Weinberg et al. (2011) are 1.0 pg µL$^{-1}$ and 10 pg µL$^{-1}$, respectively. While the LOD and LOQ values for PFOS in our study are 0.23 pg mL$^{-1}$ and 0.71 pg mL$^{-1}$, respectively.

To clarify this, we have added the following text to the Section 3.2 of the main text: "*It is important to note that the later study considered the same set of ionic PFAS as our study but included two additional analytes i.e. PFDoA and perfluorosulfonamide (PFOSA), which were not targeted by our method.*" (lines 303, page 12 of the revised main text).

Lines 301 – 302 – add in the polyfluoralkylphosphate esters – See Ao et al., 2024 (DOI provided in previous comment).

**Response:** We thank the reviewer for this comment. The *polyfluoralkylphosphate esters* are now mentioned in the text. (lines 315, page 13 of the revised main text).

SI Tables S3 and S4 – list the CAS number for each compound

**Response:** The CAS number of each compound listed in Tables S3 and S4 has been now included in Tables S1.

SI Tables S4 and S5 – I assume the reported SD is the SD for replicate injections? Please state in this tables. I also recommend replacing CSB with the measured value as this is more informative, especially if the authors include the blank concentrations, LOD, and LOQ values in the SI.

**Response:** Thank you for this comment. This has now been addressed in the captions of Tables S4 and S5 of SI: "*The reported standard deviation of the concentrations is from three replicate injections.*" (line 58 and 70, page 4 and 5 of the revised SI).

The system blank values are now included in the SI in Tables S6 and S7.

Technical Corrections:

Line 43: …"PFAS that are thought to be less…" – change are to were; considering lines 45 – 47 states that studies have shown that replacement PFAS have similar adverse health effects as long chain counterparts.

**Response:** "Are" was replaced with "*were*".

Line 60: WWTP – define at first use in main text

**Response:** This has now been addressed in Section 1 of the main text.

Line 62: TSP – define at first use

**Response:** The acronym 'TSP' has been defined on its first use in line 59 of the main text in the original submission.

Line 70: change depend to depends -> "the distribution of PFAS depend[s] on the type…"

**Response:** "Depend" was replaced by "*depends*".

Line 129: change transition to partition

**Response:** "Transition" was replaced by "*partition*".

Line 281: change size to diameter

**Response:** "Size" was replaced by a "*diameter*".

**References**

Kourtchev, I., Hellebust, S., Heffernan, E., Wenger, J., Towers, S., Diapouli, E., and Eleftheriadis, K.: A new on-line SPE LC-HRMS method for the analysis of Perfluoroalkyl and Polyfluoroalkyl Substances (PFAS) in $PM_{2.5}$ and its application for screening atmospheric particulates from Dublin and Enniscorthy, Ireland, Sci. Total Environ., 835, 155496, https://doi.org/10.1016/j.scitotenv.2022.155496, 2022.

Pubchem Perfluorobutanesulfonic acid. https://pubchem.ncbi.nlm.nih.gov/compound/Perfluorobutanesulfonic-acid, last access: 26 February 2025.

Sha, B., Ungerovich, E., Salter, M. E., Cousins, I. T., and Johansson, J. H.: Enrichment of Perfluoroalkyl Acids on Sea Spray Aerosol in Laboratory Experiments: The Role of Dissolved Organic Matter, Air Entrainment Rate and Inorganic Ion Composition, Environ. Sci. Technol. Lett., 11, 746-751, https://doi.org/10.1021/acs.estlett.4c00287, 2024.

Weinberg, I., Dreyer, A., and Ebinghaus, R.: Waste water treatment plants as sources of polyfluorinated compounds, polybrominated diphenyl ethers and musk fragrances to ambient air, Environ. Pollut., 159, 125-132, https://doi.org/10.1016/j.envpol.2010.09.023, 2011.

Zhang, M. and Suuberg, E. M.: Estimation of vapor pressures of perfluoroalkyl substances (PFAS) using COSMOtherm, J. Hazard. Mater., 443, 130185, https://doi.org/10.1016/j.jhazmat.2022.130185, 2023.

---

## Author Comment (AC2)

**Comment:** In their manuscript, the authors present results from collecting PM10 samples from a scaled-down activated sludge treatment tank and analyzing the samples for several emerging and legacy PFAS. The study was conducted in October 2023 and repeated in March 2024 in the UK, and domestic sludge from a regional WWTP was used. Overall, the manuscript is well written and appropriately structured, and it fills a gap in the literature regarding the study of aerosolization of PFAS during WW treatment. However, there are a few issues that the authors should address.

**Response:** We would like to sincerely thank Reviewer 1 for his/her important comments and for recognising the significance of our study in addressing the knowledge gap regarding the aerosolisation of PFAS during wastewater treatment. We appreciate the reviewer's acknowledgment that the manuscript is well-written, appropriately structured, and fills the gaps in the literature. Our detailed responses to the reviewer's comments are provided below in blue. The new text added to the revised manuscript is shown below in italics.

Major issues:

- From the brief description in the abstract and the slightly more extensive description in the Methods section (L. 107-112), the setup of the scaled down (what is the scale?) in conjunction with the "parent WWTP" is unclear and I strongly suggest adding more detail, including the flowrate of the sludge that is supplied to the smaller tank and how it is connected to the main WWTP.

    **Response:** The authors thank the reviewer for the comment. We have a strict non-disclosure agreement (NDA) with the wastewater treatment company where we collected the samples. The NDA imposes restrictions on us to add details such as the scale of the smaller tank in comparison to the main tanks, the flow rate of the sludge, and how the scaled-down tank is connected to the main WWTP. However, we strongly believe that even without this information, our results still provide critical information on PFAS emissions, particularly for legacy PFAS that remain in wastewater streams despite their ban, highlighting their persistence in sewage processing and their distribution beyond expected environments, including partitioning between water and aerosol phases in sewage processing.

    In the conclusion of the abstract and also in the Results & Discussion section, the concerns associated with aerosolization of PFAS from WWTP remains vague and should be summarized more explicitly, given the persistence of PFAS in the environment and their ability to travel long distances. As other (industrial) sources become more restricted, the focus likely has to shift to sources like WWTPs. Please add this discussion to the manuscript.

    **Response:** We sincerely appreciate the reviewer's insightful comment. However, our study primarily focuses on quantifying airborne PFAS concentrations within the WWTP atmosphere and was therefore submitted as a measurement report. The study aims to demonstrate that WWTP processes have the potential to introduce PFAS into the atmosphere, even when treating domestic wastewater. A comprehensive assessment, including concurrent measurements of PFAS levels in both wastewater and air, would be necessary to fully characterise the extent and discuss the implications of PFAS aerosolisation from WWTPs.

    We believe that our study serves as a foundation for future research on PFAS emissions from domestic waste sources. Future studies should incorporate simultaneous monitoring of PFAS

in wastewater, particulate matter, and the gaseous phase to enhance our understanding of emission pathways and associated risks.

We have now added a limitation section to address the above comment: "*Future research should consider simultaneous characterisation of wastewater PFAS levels alongside PM measurements to improve understanding of the relationship between airborne PFAS emissions. Expanding the range of monitored PFAS beyond the 15 fully validated targets in our study, particularly including neutral PFAS such as FTOHs and FOSEs, would enhance understanding their role in the WWTP aerosolisation. Additionally, incorporating gas-phase sampling would be valuable in assessing the potential partitioning of PFAS into the gaseous phase, further refining our understanding of their atmospheric behaviour.*" (lines 326-331 page 13 of the revised main text).

Nonetheless, as stated above, our findings provide critical insights into how domestic wastewater treatment can facilitate PFAS distribution in unexpected environmental compartments, particularly through partitioning between water and aerosol phases during sewage processing.

- In conjunction with the previous comment, in L. 83 the authors state "...a knowledge gap exists regarding the atmospheric fate of PFAS...", which I suggest rephrasing as "...a knowledge gap exists regarding the release of PFAS into the atmosphere..." to better fit the context of this manuscript.

  **Response:** As suggested, the original text was changed to "*Consequently, a knowledge gap exists regarding the release of PFAS to the atmosphere, particularly their association with $PM_{10}$ aerosols, during the treatment of domestic wastewater, especially under conditions of vigorous aeration.*" (lines 83-85, page 3 of the revised main text).

- Please provide the LODs and LOQs for this study in the SI.

  **Response:** The LODs and LOQs for this study have been included in the SI as Table S5.

- As part of the discussion in L. 179-189 it should be stated that PFAS levels were not measured in the sludge itself, which makes any explanation for the observed trends highly speculative. This fact is touched upon in L. 222-224 and again in L. 246-251, but needs to be mentioned earlier and more explicitly.

  **Response:** As suggested, the following sentence has been added to the Section 3.1 of the main text: "*It should be noted that the concentrations of PFAS in the wastewater were not measured in our study.*" (line 201-202, page 9 of the revised main text). This is also now mentioned in the limitation section: "*Future research should consider simultaneous characterisation of wastewater PFAS levels alongside PM measurements to improve understanding of the relationship between airborne PFAS emissions. ….*" (lines 326-331 page 13 of the revised main text).

- I suggest moving the three sentences from L. 246-251 to L. 181, because the discussion provided in these sentences is important for context at this earlier point.

  **Response:** We thank the reviewer for this comment. This has been now addressed in Section 3.1 of the main text: "PFBA is one of the most volatile PFAS observed in our study. Further, short chain PFAS such as PFBA and PFBS are reported to have lower aerosolisation tendencies compared to long chain compounds (e.g., PFOS and PFOA) (Johansson et al.,

2019; Pandamkulangara Kizhakkethil et al., 2024). Despite being volatile and having low aerosolisation tendency, the presence of PFBA in the $PM_{10}$ aerosols at considerable concentrations in our study could potentially be due to the presence of high levels of PFBA in the sewage during the sampling period." (lines 188-193, page 9 of the revised main text).

- 190-200: The authors are discussing different general sources of the detected PFAS, but are not clear about how these PFAS might have been transferred into the sewage sludge. Are the authors aware of any of those industries in their sampling region? What about levels of PFAS in the local drinking water supply, which likely makes up a large portion of the wastewater? Laundry water may also be a source, as PFAS have been detected in dryer lint.

  **Response:** As detailed in the manuscript, the WWTP treats domestic wastewater, with no known nearby industries that could serve as a potential point source of PFAS. Regarding the drinking water as source of PFAS, we are not clear about this question. We agree with the reviewer that laundry water may be one of the many sources of PFAS in the domestic wastewater. This has been now included in Section 3.1 of the main text: "*Laundry water could potentially be one of the sources of PFAS in the sewage since the WWTP receive a major portion of the sewage from households* (Clara et al., 2008)." (lines 213-214, page 9 of the revised main text).

- 225-232: The discussion of the diurnal trends seems incomplete: The only clearly observable trend appears to be that a larger number of PFAS are present above LOD/LOQ during the night compared to the day. Why is that? If a compound is detected during the day, its level is ballpark similar to the corresponding night measurement. PFOA appears to be the only exception with clear spikes on some days. Maybe people do more of their laundry in the evening, run the dishwasher, produce more PFAS containing WW? Please revise this section.

  **Response:** Thank you for this comment. The sewage in the AS tank is mixed and contains wastewater from both the day and the night-time periods. As detailed in Section 2.2, the scaled-down tank receives and processes sewage from the main tanks, which is already primary treated. Therefore, the influence of diurnal household activities could not be captured in our samples. One of the potential reasons for the diurnal variations of PFAS concentrations could be the shorter sampling duration of the daytime samples (collected between 10.00 AM-3.00 PM) in comparison that of to the night samples (collected between 3.00 PM and 10.00 AM the following day). The MiniVol™ tactical air sampler does not allow automatic filter change requiring us to manually replace filters within the facility's access hours. The shorter sampling duration of day samples could lead to lower PFAS mass load in the samples and thereby <LOD values. This information has been now included in Section 3.1: "*The shorter sampling duration of day samples compared to the night samples likely led to a lower aerosol mass load on the filters, resulting in several PFAS mass loads below the LODs, which could explain the observed diurnal differences in PFAS concentrations.*" (lines 248-250, page 10 and 11 of the revised main text).

- I recommend adding a Limitations section, which should include at a minimum the following considerations: only one location was sampled during limited time frame, the sewage sludge was not analyzed to investigate the relationship between PFAS concentrations in PM10 and in the sludge, PM10 concentrations were not measured nor was the PM10 further characterized, and neutral PFAS (e.g., FTOHs, FOSEs) were not included in the analysis.

**Response:** A limitation section has been now included in the main text: "*Future research should consider simultaneous characterisation of wastewater PFAS levels alongside PM measurements to improve understanding of the relationship between airborne PFAS emissions. Expanding the range of monitored PFAS beyond the 15 fully validated targets in our study, particularly including neutral PFAS such as FTOHs and FOSEs, would enhance understanding their role in the WWTP aerosolisation. Additionally, incorporating gas-phase sampling would be valuable in assessing the potential partitioning of PFAS into the gaseous phase, further refining our understanding of their atmospheric behaviour.*" (lines 326-331 page 13 of the revised main text).

Minor issues:

- I understand that this manuscript has been submitted as a measurement report, but does that categorization have to be part of the title? The "Measurement report" part is missing from the title in the SI, so at least the title should be made consistent.

  **Response:** The requirement of the journal for SI title is as follows "**Supplements will receive a title page added during the publication process including title ("Supplement of"), authors, and the correspondence email. Therefore, please avoid providing this information in the supplement**". Therefore, the title of the SI has been now removed.

- Throughout the manuscript, I suggest replacing several instances of "aerodynamic size" with "aerodynamic diameter", including in the abstract.

  **Response:** This has been now addressed in the main text (lines 14, 70, 280, 281, and 294 in pages 1,3, 11, 12, and 12, respectively, of the revised main text).

- 114: Define "near" – how far from the aeration tank was the sampler installed?

  **Response:** The sampler was placed at a distance less than 0.2 m from the aeration tank. This information has been now included in Section 2.3 of the main text: "The MiniVol™ tactical air sampler (Air Metrics, United States of America) used for $PM_{10}$ sampling was installed near the aeration tank *(<0.2 m from the aeration tank) with the sampling head slightly above the rim of the tank (10 cm above).*" (lines 114-115, page 4 of the revised main text).

- 118: Define "day" and "night" – what were the time frames? And why was sampling so much shorter during the day compared to the night?

  **Response:** We thank the reviewer for this important comment. Day sampling started after 10:00 AM and finished before 3:00 PM, while night sampling began after 3:00 PM and ended before 10:00 AM the following morning. As mentioned in our response to another question above, the MiniVol air sampler does not allow for automatic filter changes at a specified time and thus filters were manually changed during the allowed access hours to the facilities. Due to these limitations, the sampling duration for the daytime samples was shorter than that of the nighttime samples.

  The 'day' and 'night' have been now defined in Section 2.3 of the main text: "*$PM_{10}$ samples were collected separately during day (between 10.00 AM and 3.00 PM) and night (between 3.00 PM and 10.00 AM the next day).*" (lines 118-119, page 4 of the revised main text).

- 128: Regarding the importance of sampling artifacts, especially for PFAS, please consider Chang et al. (2024), "Indoor air concentrations of PM2.5 quartz fiber filter-collected ionic

PFAS and emissions to outdoor air: findings from the IPA campaign"
(https://doi.org/10.1039/D4EM00359D).

**Response:** The suggested reference is now cited in Section 2.3 of the main text "It is important to note that the use of GFF and quartz fiber filters (QFF) during PM sampling has been reported to cause positive sampling artefacts, such as the adsorption of gas-phase organic compounds (*Chang et al., 2025*; Turpin et al., 1994)." (lines 129-130, page 5 of the revised main text).

- 131: Delete "slightly".

  **Response:** The word 'slightly' has been removed.

- 139: What were the filters prewashed with?

  **Response:** The filters were prewashed with 15 mL of methanol as described in Kourtchev et al. (2022). This has been now addressed in Section 2.4 of the main text: "*The methanol extracts were then filtered through a prewashed with methanol (3 times with 5 mL) 0.45 μm PTFE syringe filter.*" (line 140-141 page 6 of the revised main text).

- Tables S3 and S4 mention that the results were blank corrected. Please add a brief description about the process to the Methods section.

  **Response:** We thank the reviewer for this important comment. This has now been included in Section 2.4 of the main text: "*The maximum concentration value of the PFAS detected in the field blanks and baked filter blanks were subtracted from the PFAS concentrations detected in the samples.*" (lines 163-165, page 6 of the revised main text).

- Section 2.5: Were field blanks collected?

  **Response:** Field blanks were collected during the campaign. This was mentioned in Section 2.3 line 125 of the main text in the original submission. For more clarity this has now been modified to "*Two types of blanks were used to evaluate possible PFAS contamination from handling the filters. These include: 1) baked filters (BF) and 2) baked filters placed in MiniVol® air sampler and collecting air above the AS tank at 10 L min$^{-1}$ for 2 min (field blanks, FLDB).*" (lines 125-128, page 5 of the revised main text).

  The blank values including the field blank values have now been added to the SI as Tables S6 and S7.

- 208-209: The sentence about the pH value is difficult to understand. Please rephrase.

  **Response:** The sentence was revised to "*The pH of the contaminated water has been reported to influence the atmospheric transfer of PFAS*". (lines 222-223, page 10 of the revised main text).

- 218: Period missing after "concentration".

  **Response:** the period (.) has been added.

- 218-219: Rain water may also dilute the wastewater, depending on the PFAS sources.

  **Response:** This has been now included in Section 3.1 of the main text: "*Additionally, dilution of PFAS levels in the sewage due to rainfall could also affect the airborne PFAS concentration.*" (lines 233-234, page 10 of the revised main text).

- 252 and L. 257: It is my understanding that Weinberg et al. measured PFAS in TSP and not "estimated" the concentrations. "Estimated" indicates theoretical/modeling results. Please revise.

  **Response:** The word 'estimated' is now replaced with '*measured*'.

- 261: Why the parentheses around "up to 1.31 pg m-3"?

  **Response:** The parentheses have been removed.

- 307: Replace "could potentially represent" with "likely represent".

  **Response:** Done.

- 320: Replace "filed" with "field".

  **Response:** "filed" has now been replaced with "*field*".

- SI: I recommend including the full author list in the SI instead of using "Jishnu Pandamkulangara Kizhakkethil et al.".

  **Response:** The requirement of the journal for SI title is as follows "**Supplements will receive a title page added during the publication process including title ("Supplement of"), authors, and the correspondence email. Therefore, please avoid providing this information in the supplement.**". Hence, the author information has been now removed from SI.

- SI: Please include table and figure captions in the table of content.

  **Response:** The table and figure captions have been included in the table of contents of the SI.

- SI: Please list CAS RNs or other unique identifiers in Table S1 together with the PFAS name and abbreviation.

  **Response:** The CAS RNs for the PFAS in the EPA 533 PAR mix have now been included in Table S1 of SI.

- SI: Please indicate clearly in the captions of Tables S3 and S4 that none of the other targeted PFAS were detected in any sample. If any were detected at least once, I recommend including the concentration in the provided tables.

  **Response:** We thank the reviewer for this comment. This has now been addressed in the captions of Tables S3 and S4 of SI: "*Other targeted PFAS amenable to the method but not listed in the table were either below the method's LOD or not detected in any of the collected samples*". (lines 62 and 76, page 4 and 5, of revised SI).

**References**

Clara, M., Scharf, S., Weiss, S., Gans, O., and Scheffknecht, C.: Emissions of perfluorinated alkylated substances (PFAS) from point sources—identification of relevant branches, Water Sci. Technol., 58, 59-66, https://doi.org/10.2166/wst.2008.641, 2008.

Weinberg, I., Dreyer, A., and Ebinghaus, R.: Waste water treatment plants as sources of polyfluorinated compounds, polybrominated diphenyl ethers and musk fragrances to ambient air, Environ. Pollut., 159, 125-132, https://doi.org/10.1016/j.envpol.2010.09.023, 2011.